# TDP-43 Controls HIV-1 Viral Production and Virus Infectiveness

**DOI:** 10.3390/ijms24087658

**Published:** 2023-04-21

**Authors:** Romina Cabrera-Rodríguez, Silvia Pérez-Yanes, Iria Lorenzo-Sánchez, Judith Estévez-Herrera, Jonay García-Luis, Rodrigo Trujillo-González, Agustín Valenzuela-Fernández

**Affiliations:** 1Laboratorio “Inmunología Celular y Viral”, Unidad de Farmacología, Sección de Medicina, Facultad de Ciencias de la Salud, Universidad de La Laguna (ULL), 38320 La Laguna, Tenerife, Spain; sperezya@ull.edu.es (S.P.-Y.); alu0101119878@ull.edu.es (I.L.-S.); jesteveh@ull.edu.es (J.E.-H.); jgarcial@ull.edu.es (J.G.-L.); rotrujil@ull.edu.es (R.T.-G.); 2Analysis Department, Faculty of Mathematics, Universidad de La Laguna (ULL), 38296 La Laguna, Tenerife, Spain

**Keywords:** TARDBP/TDP-43, HDAC6, autophagy, viral production, HIV-1 infection capacity

## Abstract

The transactive response DNA-binding protein (TARDBP/TDP-43) is known to stabilize the anti-HIV-1 factor, histone deacetylase 6 (HDAC6). TDP-43 has been reported to determine cell permissivity to HIV-1 fusion and infection acting on tubulin-deacetylase HDAC6. Here, we studied the functional involvement of TDP-43 in the late stages of the HIV-1 viral cycle. The overexpression of TDP-43, in virus-producing cells, stabilized HDAC6 (i.e., mRNA and protein) and triggered the autophagic clearance of HIV-1 Pr55^Gag^ and Vif proteins. These events inhibited viral particle production and impaired virion infectiveness, observing a reduction in the amount of Pr55^Gag^ and Vif proteins incorporated into virions. A nuclear localization signal (NLS)-TDP-43 mutant was not able to control HIV-1 viral production and infection. Likewise, specific TDP-43-knockdown reduced HDAC6 expression (i.e., mRNA and protein) and increased the expression level of HIV-1 Vif and Pr55^Gag^ proteins and α-tubulin acetylation. Thus, TDP-43 silencing favored virion production and enhanced virus infectious capacity, thereby increasing the amount of Vif and Pr55^Gag^ proteins incorporated into virions. Noteworthy, there was a direct relationship between the content of Vif and Pr55^Gag^ proteins in virions and their infection capacity. Therefore, for TDP-43, the TDP-43/HDAC6 axis could be considered a key factor to control HIV-1 viral production and virus infectiveness.

## 1. Introduction

The transactive response DNA-binding protein (TARDBP; also known as transactive response DNA binding protein 43 kDa (TDP-43)) is a mainly nuclear RNA-binding protein that carries out very important biological tasks in cells such as splicing, transcription and translation, mRNA transport, mRNA stability, and pri-miRNA processing [1,2,3,4,5,6,7,8,9,10]. Likewise, TDP-43 is able to interact with and influence from hundreds to thousands of transcripts, with the histone deacetylase enzyme 6 (HDAC6) being one of those transcripts highly regulated by TDP-43 [11,12]. Thus, the level of expression of the TDP-43 protein directly affects the mRNA and protein levels of HDAC6 [11,12]. HDAC6 is an enzyme that promotes the deacetylation of the α-tubulin subunit in microtubules (MTs), thereby modulating cytoskeleton dynamics [13,14,15,16,17,18] and the first steps of the HIV-1 infection process [15,19,20], as well as the regulation of HIV-1 replication and infectious ability by promoting the autophagy-mediated clearance of several viral factors (e.g., Pr55^Gag^ and Vif) [21,22]. Notably, TDP-43 establishes the permissive status of cells to be infected by HIV-1 through the regulation of HDAC6 mRNA and protein expression levels and its aforementioned deacetylation activity [12]. The relevance of this new TDP-43 function has been demonstrated by using HIV-1 viral particles carrying primary viral envelopes (Envs) isolated from viruses of HIV-1-infected individuals with different clinical outcomes. Thus, the overexpression of TDP-43 inhibits the infection activity of functional Envs from viruses of viremic non-progressor (VNP) and rapid progressor (RP) patients down to the levels of inefficient HIV-1 Envs from viruses of long-term non-progressor elite controller (LTNP-EC) individuals [12,19,23]. In fact, TDP-43 modulates the infectiveness of functional HIV-1 Envs that could induce late autophagy in primary non-infected CD4+ T cells, causing death by the contact of these bystander cells, in an Env/CD4-dependent manner [23,24]. On the contrary, the specific silencing of endogenous TDP-43 significantly enhances the infectious capacity of primary Envs isolated from the HIV-1 virus of VNP and RP patients, even restoring the infection capacity of inefficient Envs isolated from viruses of LTNP-EC individuals [12]. These TDP-43 effects on HIV-1-Env-mediated pore fusion formation and infection are exerted through the antiviral factor HDAC6 in a tubulin-deacetylase manner [12]. Therefore, the ability of HIV-1 Envs to overcome the HDAC6 barrier (i.e., the TDP-43/HDAC6 axis) and subsequently stabilize cortical MTs conditions the efficiency of the early steps of the HIV-1 infection cycle [12,20,23,25].

Furthermore, HDAC6-mediated autophagy is involved in the control of the viral production and virion infection capacity of HIV-1 [21,22]. Thus, HDAC6 forms a complex with the Apolipoprotein B mRNA-editing enzyme catalytic polypeptide-like 3 (A3G), impeding the proteasomal degradation of A3G by the viral protein Vif and interacting with the Vif factor to promote its cellular clearance by autophagy [22]. In this regard, HDAC6 inhibits HIV-1 production by targeting the viral polyprotein Pr55^Gag^ by autophagic degradation [21,22]. This HDAC6 anti-HIV-1 activity is counteracted by the HIV-1 Nef factor that associates with the deacetylase, pro-autophagy enzyme [21]. HIV-1 Nef targets HDAC6 by acidic/endosomal-lysosomal processing, stabilizing Pr55^Gag^ and Vif viral proteins, thus assuring Pr55^Gag^ localization and aggregation at the plasma membrane, viral egress, and the infectivity of viral particles (i.e., by stabilizing the HIV-1 Vif protein) [21]. Altogether, these data indicate that HDAC6 is a key restriction factor that enhances its expression levels or activity (i.e., antiviral MTs deacetylation and autophagy) and could be a good strategy to explore against HIV-1 infection.

Considering all the above presented information, in this work, we aimed to determine the functional involvement of TDP-43 (i.e., the TDP-43/HDAC6 axis) in the late stages of the HIV-1 life cycle by studying their effects on the efficiency of viral particle production and virion infection capacity. Our results indicated that the TDP-43/HDAC6 axis can regulate HIV-1 replication acting on Pr55^Gag^ and Vif viral proteins to control viral particle production and virion infectiveness.

## 2. Results

### 2.1. TDP-43 Expression and Its Effect on HDAC6 Expresion

To ascertain the effect of TDP-43 on HIV-1 viral production and virion infection capacity, we first studied the expression and effect of TDP-43 on HDAC6 in a well-accepted cell model for studying the HIV-1 infection cycle, such as viral particle production, as previously reported [21,22]. We characterized the effect of two TDP-3 constructs that have previously been assayed on HIV-1 infection studies [12] and well characterized in other biological functions [11,12,26,27,28]: a full-length N-terminal Flag-tagged, wild-type TDP-43 construct (Flag-wt-TDP-43), and an N-terminal Flag-tagged nuclear localization signal (NLS) mutant (Flag-NLS-mut-TDP-43) that lacks the NLS domain of the protein [11,12,26,27,28], as we previously characterized in HIV-1 studies [12]. We observed that the overexpression of the full-length TDP-43 construct increases and stabilizes the level of expression of the tubulin-deacetylase HDAC6 (Figure 1A, western blot line 2, quantified in the associated histograms on the right), which provokes a decrease in the levels of acetylated α-tubulin in stable MTs (Figure 1A, western blot line 2, quantified in the associated histograms on the right), a main substrate of the enzyme. The over-expression of the NLS-mut-TDP-43 construct did not affect the level of expression of the HDAC6 enzyme, and no significant changes were seen in the levels of acetylated α-tubulin on MTs (Figure 1A, western blot line 3, quantified in the associated histograms on the right).

These data are in accordance with our reported biochemical characterization of these constructs [12]. Furthermore, the overexpression of the two TDP-43 constructs increased the level of mRNA of TDP-43, as quantified by reverse transcription quantitative real-time PCR (RT-qPCR) (Figure 1B, left histogram). Noteworthy, the TDP-43 functional construct had a significant effect on HDAC6 mRNA levels, stabilizing higher levels compared to those observed in the control, which were untreated HEK-293T cells (Figure 1B, right histogram). NLS-mut-TDP-43 had no effect on the levels of the HDAC6 mRNA (Figure 1B, right histogram), as previously reported [12].

Altogether, these data corroborate the function of TDP-43 in stabilizing HDAC6 mRNA and protein in HEK-293T cells (Figure 1).

### 2.2. TDP-43-Mediated HDAC6 Protein Stabilization Promotes Degradation of the HIV-1 Pr55^Gag^ and Vif Viral Proteins: Impairing Viral Particle Production and Virus Infection Capacity

Next, we analyzed the effect of the TDP-43-mediated HDAC6 protein increase on HIV-1 viral production and the infection capacity of released viral particles. Therefore, we overexpressed TDP-43 constructs in HEK-293T cells producing HIV-1 virions (Figure 2). As presented above, the overexpression of the full-length TDP-43 construct stabilized the level of expression of HDAC6 (Figure 2A,B). This wt-TDP-43 action on HDAC6 directly correlates with the clearance of the viral proteins Vif and Pr55^Gag^ (Figure 2A, biochemical analysis, lines 3–6, with band intensity quantification in the histograms on the right, and Figure 2B, western blot line 3 associated bands) as reported for the anti-HIV-1 action of HDAC6 [21,22], observing a TDP-43 dose-response degradative effect on these two viral proteins (Figure 2A). Moreover, a correlation was observed between the stabilization of the HDAC6 enzyme and the clearance of these two viral proteins, as well as for the marker of autophagy activity, SQSTM1 (sequestosome 1) or p62 (Figure 2A, see associated western blot band). The p62 protein connects ubiquitinated protein aggregates to autophagosomes and facilitates their clearance [29,30]. Likewise, this effect of TDP-43 on p62 is dose-response dependent (Figure 2A). These data are in accordance with previous reports indicating that HDAC6 degrades Vif and Pr55^Gag^ by autophagy with the concomitant decay of p62 [21,22]. Furthermore, we observed that HDAC6 could not act on its viral targets, Vif and Pr55^Gag^, in cells overexpressing the NLS-TDP-43 mutant (Figure 2B, western blot line 4, quantified in associated right histograms), as was similarly observed with its acetylated α-tubulin substrate on MTs (Figure 1A and Figure 2B, associated line 3 and line 4 western blot bands, respectively). In fact, we reported this NLS-TDP-43 effect on HDAC6 and its MT substrate in previous studies of HIV-1 infection [12].

Therefore, this TDP-43-mediated degradation of Pr55^Gag^ (Figure 2) could account for the inhibition of viral particle production observed under this experimental condition (Figure 3A, HIV-1 viral production histogram), whereas the clearance of Vif (Figure 2) could negatively affect the infectious capacity of produced HIV-1 virions (Figure 3A, data on the HIV-1 infection capacity histogram), as previously reported for HDAC6 [21,22]. The infectious capacity of HIV-1 virions was assayed by infecting permissive CEM.NKR-CCR5 CD4+ T cells with synchronous doses of viral particles produced in HEK-293T cells overexpressing wt-TDP-43 (Figure 3A, data on the HIV-1 infection capacity histogram). This TDP-43 anti-HIV-1 effect was not observed by using the NLS-TDP-43 mutant (Figure 3A, see NLS-TDP-43 mutant associated data), which in turn had no effect on Vif and Pr55^Gag^ viral proteins (Figure 2B, western blot line 4, quantified in associated histograms on the right).

To elucidate the reason why viral particles generated in wt-TDP-43 overexpressing HEK-293T cells lost infection capacity, we precipitated viral particles from synchronous doses of cell-free supernatants (see Materials and Methods) (Figure 3B, see p24 viral control load-associated bands, lines 1–3, quantified in right histograms), and we analyzed Vif and Pr55^Gag^ protein viral content in these viral pellets and normalized p24 content as a control for synchronous viral load (Figure 3B).

We observed that the amount of both Pr55^Gag^ and Vif proteins incorporated in virions released from HEK-293T cells overexpressing wt-TDP-43 was significantly lower compared to the amount of Pr55^Gag^ and Vif proteins detected in viral particles produced in both control cells and HEK-293T cells overexpressing the NLS-TDP-43 mutant (Figure 3B, western blot lines 1–3, quantified and normalized by p24 viral content in right histograms). Thus, viral particles obtained in cells overexpressing the NLS-TDP-43 mutant were more efficient when infecting permissive CD4+ T cells compared to viral particles carrying low amounts of Vif and Pr55^Gag^ proteins (Figure 3A, data on the HIV-1 infection capacity histogram), as obtained from wt-TDP-43 overexpressing cells (Figure 3A,B) with stabilized and increased levels of HDAC6 (Figure 2A,B).

Altogether, these data indicate that TDP-43, by stabilizing HDAC6 mRNA and protein levels, negatively affects HIV-1 viral production and virus infectiveness by targeting Pr55^Gag^ and Vif viral proteins.

### 2.3. TDP-43 Degradative Action on HIV-1 Pr55^Gag^ and Vif Proteins Is Mediated by Autophagy

The results presented above indicate that the overexpression of wt-TDP-43 triggers the clearance of Pr55^Gag^ and Vif viral proteins, concomitantly with the stabilization of the proautophagy HDAC6 enzyme (Figure 1 and Figure 2). We previously reported that HDAC6 regulates HIV-1 viral production and infection by degrading Pr55^Gag^ and Vif viral proteins by autophagy [21,22]. In this matter, we analyzed the effect of the chemical inhibition of the autophagy-associated process, exerted by 3-methyladenine (3-MA) on TDP-43-mediated Pr55^Gag^ and Vif degradation, in virus-producing HEK-293T cells. The 3-MA is an inhibitor for the autophagic sequestration of cytoplasmic proteins [31,32,33] and was reported to block the HDAC6-mediated autophagy degradation of Vif and Pr55^Gag^ proteins [21,22]. Thus, in 3-MA treated cells, the blockade of the autophagic process was signaled by the stabilization of the p62 protein, as compared to control conditions (i.e., PBS vehicle control), where p62 faded (Figure 4, 3-MA vs. PBS p62-band in associated western blots). In virus-producing cells, 3-MA treatment abrogated the TDP-43-mediated degradation of the viral proteins Vif and Pr55^Gag^ (Figure 4A, western blot line 3, quantified in Figure 4B), whereas an increase of TDP-43-dependent HDAC6 was still observed (Figure 3, line 3 of the western blot analysis). This fact suggests that the inhibitory effect of 3-MA affects HDAC6-mediated viral protein degradation by autophagy, as previously reported [21,22]. We also sought the effect of the proteasome inhibitor MG132 (a synthetic peptide aldehyde) [34,35,36] on TDP-43-mediated Vif and Pr55^Gag^ clearance (Figure 5). The results obtained indicate that MG132 treatment did not affect TDP-43-mediated Vif and Pr55^Gag^ degradation (Figure 5A, western blot line 3, quantified in Figure 5B), and the stabilization of the HDAC6 enzyme was observed (Figure 5A, line 3 of the DMSO and MG132 western blot analysis). These data are in accordance with previously reported data that point to the fact that that proteasome inhibition by MG132 does not affect HDAC6-mediated Vif and Pr55^Gag^ degradation [21,22]. Moreover, we observed that the NLS-TDP-43 mutant did not affect Vif and Pr55^Gag^ degradation under this (Figure 5) and related experimental conditions (Figure 2 and Figure 4).

Altogether, using these data, autophagy appears as a central process in the HIV-1 antiviral activity of the TDP-43/HDAC6 axis, thereby promoting the degradation of the Pr55^Gag^ and Vif viral proteins. These events resulted in the inhibition of HIV-1 viral production and in the inefficient capacity of nascent virions to infect.

### 2.4. Specific TDP-43 Knockdown Diminishes HDAC6 mRNA and Protein Levels, Favoring HIV-1 Viral Production and Virus Infection Capacity

Next, we aimed to validate the functional effect of TDP-43 on HDAC6, Pr55^Gag^, and Vif viral proteins by specific RNA interference (RNAi) using short interference RNA (siRNA) oligos against endogenous TDP-43 mRNA in virus-producing HEK-293T cells. We used two different siRNA-TDP-43 oligos (i.e., a mix of siRNA-TDP-43 (B + C) oligos (see Materials and Methods)) (Figure 6), previously characterized for efficient TDP-43 knockdown in permissive cells for HIV-1 infection [12]. The biochemical analysis of TDP-43 protein expression (Figure 6A), as well as TDP-43 mRNA levels by RT-qPCR (Figure 6B), indicated that the siRNA of endogenous TDP-43 diminished the amount of its mRNA (Figure 6B) and protein (Figure 6A). Likewise, the level of expression of endogenous HDAC6 (i.e., enzyme and mRNA) was strongly reduced in siRNA-TDP-43-treated cells (Figure 6A,B). Thus, TDP-43-mediated HDAC6 knockdown provokes an important increase in the MT acetylation of the α-tubulin subunit (Figure 6A, quantified in the associated histograms on the right).

Furthermore, specific TDP-43 knockdown leads to an increase of the expression level of the HIV-1 Vif and Pr55^Gag^ proteins in virus-producing HEK-293T cells (Figure 7A). Likewise, viral production was strongly increased under this experimental condition (Figure 7B, HIV-1 viral production histograms), thereby observing that released virions were more efficiently infecting permissive CD4+ T cells than those produced in the control, which was scrambled-treated HEK-293T cells (Figure 7B, HIV-1 infection capacity histograms). These observations could be associated with the enhancement on the expression levels of Pr55^Gag^ and Vif (Figure 7A). In fact, the biochemical analysis of the viral content on Pr55^Gag^ and Vif proteins of synchronous doses of cell-free viral particles produced in siRNA-treated HEK-293T cells indicated that Vif and of Pr55^Gag^ are better incorporated in nascent virions in cells lacking the TDP-43 protein compared to control cells (scrambled-treated cells) (Figure 7C). These data suggest an anti-HIV-1 action of the endogenous TDP-43 (i.e., TDP-43/HDAC6 axis) in control conditions, which is lost after TDP-43 silencing, leading to a concomitant diminution of the antiviral HDAC6 enzyme and a subsequent increase of the HIV-1 Vif and Pr55^Gag^ proteins (Figure 7A).

Altogether, these data prompted the suggestion for a new function for TDP-43 in regulating HIV-1 viral production and virion infection efficiency through destabilizing HDAC6 (i.e., mRNA and protein levels) and triggering the autophagic degradation of HIV-1 Vif and Pr55^Gag^ proteins. These degradative events impair the production of viral particles and reduce the infectious capacity of nascent virions (Figure 8, a scheme summarizing TDP-43 action on HIV-1 viral production and virion infectiveness).

## 3. Discussion

In this work, we aimed to determine the functional involvement of TDP-43 in HIV-1 viral particle production and virion infection capacity. Our results indicated that TDP-43 regulates HIV-1 replication acting on Pr55^Gag^ and Vif viral proteins. The overexpression of the full-length construct (wt-TDP-43) stabilizes and increases the mRNA and protein levels of the cellular factor HDAC6, as previously reported in permissive cells [12]. Our data indicated that the stabilization of HDAC6 in virion-producing cells overexpressing TDP-43 decreased the acetylation levels of the α-tubulin subunit of MTs. Likewise, TDP-43 has been reported to regulate cell permissivity to HIV-1 fusion and infection by acting on HDAC6, thereby modulating the levels of acetylated MTs during early steps of the viral cycle. Thus, HIV-1 Env signaling through the receptor CD4+ leads to the reorganization and stabilization of MTs, bypassing the antiviral barrier that imposes the tubulin-deacetylase HDAC6 [15,19,20,21,23,24,25]. This means that functional HIV-1 Envs with the ability to vault the HDAC6 barrier establish a productive infection, while HIV-1 Envs from viruses that have lost this ability do not set a correct signal and infection, as occurs with viral Envs isolated from viruses of EC individuals [12,19,23]. Then, HDAC6 stabilization by the overexpression of wt-TDP-43 generates a non-permissive state for target cells by impairing MT stabilization and making HIV-1 Env/CD4-mediated pore fusion formation and subsequent viral entry and infection difficult [12]. Furthermore, as observed in the present work, MTs are stabilized after TDP-43 silencing, as indicated by the enhancement of α-tubulin acetylation, generating a permissive state for HIV-1 infection and replication, as previously reported [12,15,19,20,21,23,24,25]. The nuclear localization signal (NLS)-TDP-43 mutant was not able to control the HIV-1 viral production and infection and could not stabilize HDAC6, as previously reported in CD4+ T permissive cells [12]. In this context, anti-viral HDAC6 action is not promoted, as previously reported to restrict HIV-1 infection [12,15,19,20,21,22,23]. Altogether, these data and the present study prompt the suggestion that TDP-43 and the TDP-43/HDAC6 axis are key to control HIV-1 infection.

Moreover, this TDP-43-associated increase of the pro-autophagic HDAC6 factor drives the cell to an antiviral state that leads to the clearance of the viral proteins Pr55^Gag^ and Vif. In fact, the chemical inhibition of the aggresomal formation by 3-MA impairs the TDP-43-mediated autophagic degradation of HIV-1 Pr55^Gag^ and Vif proteins. However, in the presence of a chemical inhibitor of the proteasome pathway, MG132, the overexpression of wt-TDP-43 still triggered Pr55^Gag^ and Vif clearance. Likewise, in the presence of these chemical inhibitors, TDP-43 overexpression enhanced the level of expression of the pro-autophagic HDAC6 enzyme. This means that the HDAC6 autophagy-degradative pathway could be promoted, being only blocked by the 3-MA inhibitor. Furthermore, these data are in accordance with previous reported works that demonstrated Vif and Pr55^Gag^ autophagic degradation mediated by HDAC6 [21,22]. Therefore, HDAC6-associated autophagy appears to be a central antiviral action of this enzyme [12,19,21,22,23,24], where the TDP-43/HDAC6 axis could regulate HIV-1 Vif and Pr55^Gag^ protein stability to control viral particle production and virion infectiveness. Noteworthy, the specific silencing of the endogenous TDP-43 validates the antiviral function of TDP-43 (i.e., the TDP-43/HDAC6 axis), since the TDP-43 knockdown that reduces HDAC6 expression (i.e., mRNA and protein) strongly increases the level of expression of the HIV-1 viral proteins Vif and Pr55^Gag^ in virus-producing cells as well as in released viral particles. Our results therefore indicated a clear-cut correlation between the stability of the HIV-1 Pr55^Gag^ and Vif viral proteins and the efficiency of viral production and virion infectiveness. Thus, the overexpression of wt-TDP-43 inhibits HIV-1 viral particle production and strongly reduces the infection capacity of released virions, which contain low amounts of Pr55^Gag^ and Vif viral proteins. In contrast, TDP-43 silencing reduces HDAC6 expression and significantly favors viral particle production and the infectious capacity of HIV-1 virions, correlating these facts with higher amounts of Pr55^Gag^ and Vif viral proteins in HIV-1-released virions. These data correlate with the fact that Vif has been reported to be a critical factor for HIV-1 infectiveness, as its decrease or elimination makes viral particles less infectious [21,22,37]. Of note, the NLS-TDP-43 mutant failed to control HIV-1 viral production and infection, since this construct had no effect on the levels of expression of the viral Vif and Pr55^Gag^ factors in virion producing cells.

TDP-43 was first reported as a novel protein that binds to transactivator-responsive DNA sequences within HIV-1, acting as a transcriptional repressor [38]. However, it has been proposed that TDP-43 has no effect on viral LTR transcription and virus production, as in macrophages [39]. In this work, the effect of TDP-43 on p24 release from macrophages was assayed by silencing TDP-43 24 h after infection of Vpx-treated macrophages by HIV/VSV-G-dependent endocytic uptake. Notably, it has been established that HIV-infected macrophages contain large intracellular pools of infectious virus [40,41,42], which can be released in an exosome-dependent pathway [43] or from late endosomes, where the majority of infectious HIV viral particles assemble and egress [44]. Therefore, it is plausible that under Vpx/VSV-G-mediated viral uptake, p24 expression could be promoted and released from these macrophagic compartments, preventing antiviral TDP-43 action. In permissive non-myeloid cells, HIV-1 viral particles mainly assemble at the cell surface [45,46,47,48,49,50], as we also reported in virus-producing HEK-293T cells by total internal reflection fluorescence microscopy (TIRFM) [21] and characterized the mechanism of HIV-1 infection promoted by Vif or Nef viral factors and the antiviral action exerted by HDAC6 and/or TDP-43 (TDP-43/HDAC6 axis) in this and other permissive cells [12,19,20,21,22,23,25]. Thus, these data suggest that TDP-43 controls HIV-1 viral production.

Moreover, it has been reported that TDP-43 overexpression reduces the release of the HIV-1 p24 antigen in HeLa cells co-transduced with HIV-1 pro-viral DNA [38]. The authors used an in vitro transcription assay to observe that in the presence of recombinant TDP-43, HIV-1-LTR DNA transcription is repressed. However, the ability of TDP-43 to impair HIV-1-LTR DNA transcription has not been analyzed in HeLa cells, for example, by quantitatively measuring pro-viral mRNA production. Although this possibility might not be ruled out, the results obtained in our work point in another direction. We observed that the overexpression of functional TDP-43 does not reduce the level of expression of HIV-1 Pr55^Gag^ and Vif proteins when the autophagy-mediated clearance of Pr55^Gag^ and Vif proteins is inhibited by 3-MA treatment in virus-producing cells. Therefore, TDP-43 did not appear to affect HIV-1-LTR DNA transcription under our experimental conditions. These data suggest that the anti-HIV-1 action of TDP-43 is mainly exerted by regulating the mRNA and enzyme levels of the antiviral factor HDAC6.

In a general viral scenario, TDP-43 has been reported to play antiviral roles against different viral infections. For example, against coxsackievirus B3 (CVB3), it has been studied that TDP-43 is translocated from the nucleus to the cytoplasm through the activity of the 2A viral protease, but it also cleaves TDP-43 leading to the formation of aberrant protein aggregates, while a decrease in TDP-43 levels results in an increase in viral egress [51]. Another case is several studies of the human endogenous retrovirus-K (ERV), where it was proven that functional TDP-43 failed to regulate transcriptional ERVK expression, besides the mutant forms of TDP-43, promoted ERVK protein aggregation [52], and associated with several harmful conditions and pathologies in the central nervous system (CNS) [52,53,54,55,56,57,58]. According to these data, it was demonstrated that high ERVH-RT activity was detected along with HERKV protein accumulation in cortical neurons in patients affected by HIV-1 and was combined on the other hand with the accumulation of aberrant TDP-43 and the typical neurodegeneration in HIV-1 patients [56,57,58,59].

Recently, some studies have found that SARS-CoV-2 spike protein S1 is linked and accumulates with different proteins in the CNS, as is the case for TDP-43 [60]. Likewise, another study identified TDP-43 to form inclusions in samples taken from SARS-CoV-2-infected patients [61]. All these studies indicate that TDP-43, apart from HIV-1, plays an important role in viral infections and especially in neurodegeneration associated with those infections, as it appeared to occur with the coronavirus disease 2019 (COVID-19), in which high levels of TDP-43 were reported in the acute phase of the disease [62]. Finally, our research on the role of TDP-43 in HIV-1 infection indicated that enhancing the HDAC6-associated autophagic degradative pathway through TDP-43 regulation could be decisive against HIV-1 infection, thereby impairing viral infection and spreading.

Altogether, these data indicate a new function for TDP-43 and the TDP-43/HDAC6 axis in modulating HIV-1 viral production and virion infection efficiency through the stabilization of HDAC6 and the promotion of the autophagic degradation of HIV-1 Vif and Pr55^Gag^ proteins. Therefore, TDP-43 and the TDP-43/HDAC6 axis could be considered new targets for developing anti-HIV-1 strategies to control HIV-1 infection and replication, as well as against other types of viruses.

## 4. Materials and Methods

### 4.1. Antibodies and Reagents

Rabbit anti-HDAC6 (H-300; sc-11420), Mouse anti-p62/SQSTM1 (D-3-sc-28359), Mouse anti-p24 (24-4; sc-69728), Rabbit anti-Pr55^Gag^ (Anti-HIV1 p55 + p24 + p17) (ab63917), Mouse anti-Vif (319; sc-69731), and Polybrene (sc-134220) were obtained from Santa Cruz Biotechnology (Santa Cruz, CA, USA). The neutralizing mAb RPA-T4 directed against CD4 was obtained from eBioscience (San Diego, CA, USA). Rabbit anti-TDP-43 (Catalog Number T1705); Mouse Monoclonal anti-Flag M2 (F1804); Mabs anti-α-tubulin (T6074) and anti-acetylated α-tubulin (T7451); and secondary horseradish peroxidase (HRP)-conjugated Abs, specific for any Ab species assays, were purchased from Sigma-Aldrich (Sigma-Aldrich, St. Louis, MO, USA). Linear polyethylenimine, with an average molecular mass of 25kDa (PEI-25k), was purchased from Polyscience (Polyscience, Warrington, PA, USA). Z-Leu- Leu-Leu-al or MG132 (C2211) and 3-methyladenine (3-MA) (M9281) inhibitors were purchased from Sigma-Aldrich. Complete™ Protease Inhibitor Cocktail (11697498001) was obtained from Roche Diagnostics (GmbH, Mannheim, Germany).

### 4.2. DNA Plasmids and Viral DNA Constructs

Vectors for the expression of wild-type TDP-43 (Flag-wt-TDP-43) or mutants at the import NLS signal (Flag-NLS-mut-TDP-43) were courtesy of Dr. Thorsten Schmidt [11] and were previously characterized to study HIV-1 [12]. The pNL4-3.Luc.R-E- provirus (Δnef/Δenv), the R5-tropic BaL.01-envelope (env) glycoprotein plasmid (catalog number 6070013), and pcDNA.3.1 (-) (Cat. V790-20, Invitrogen, Waltham, MA, USA) were obtained via the NIH AIDS Research and Reference Reagent Programme (http://www.aidsreagent.org/, accessed on 4 May 2022) and were assayed as previously reported [12,19,20,21,22,23,25,63,64,65,66,67].

### 4.3. Cells

The human CEM.NKR-CCR5 permissive cell line (catalog number 4376, NIH AIDS Research and Reference Reagent Program) and the virus packaging HEK-293T cells (catalog number 103, NIH AIDS Research and Reference Reagent Program) were grown at 37 °C in a humidified atmosphere with 5% CO_2_ in RPMI 1640 medium (Lonza, Verviers, Belgium) in the case of the CEM.NKR-CCR5 cells, and in DMEM (Lonza) in the case of HEK-293T, both mediums were supplemented with 10% fetal calf serum (Lonza), 1% L-glutamine, and 1% penicillin-streptomycin antibiotics (Lonza) and were free of mycoplasma (Mycozap antibiotics, Lonza). Both cell lines were regularly split every 2–3 days. The human CEM.NKR-CCR5 permissive cell line was a reference cell-based assay for the evaluation of HIV-1 infection and neutralization by Abs [68], being suitable for the general measurement of HIV-1 infection and neutralization assays by Abs or potential new anti-HIV drugs, as we have also reported [12,19,20,21,22,23,25,63,64,65,66,67]. Cell viability was quantitatively determined by the analysis of propidium iodide uptake using flow cytometry (BD-Accuri™ C6 Plus Flow Cytometer; BD Biosciences, San Jose, CA, USA), as we reported [64], and/or by light microscopic quantitation visualizing cell trypan blue stain [69] under each experimental condition.

### 4.4. Messenger RNA Silencing

We used the below indicated short interference RNA (siRNA) oligonucleotides (oligos), specifically directed against the indicated mRNA sequences of TDP-43, in order to knockdown TDP-43 expression. Transient siRNA transfections were performed using linear polyethylenimine (PEI-25k) dissolved in 150 mM of NaCl. A mixture of PEI-25k/oligos (3:1 ratio) was gently vortexed, incubated for 20–30 min at room temperature (RT), and then added to HEK-293T cells in culture. A total of 1 µM of a commercial scrambled control oligo or a TDP-43-specific siRNA oligo was used: siRNA-TDP-43 B (5′-CACUACAAUUGAUAUCAAA[dT][dT]-3′) and siRNA-TDP-43 C (5′-GAAUCAGGGUGGAUUUGGU[dT][dT]-3′) (ordered to Sigma). These siRNA-TDP-43 oligos were previously characterized [11,12]. The siRNA for TDP-43 induced the specific interference of protein expression for at least 48 h, as witnessed in western blots. Cells treated with this mixture of PEI-25k/oligos (scrambled or specific siRNA oligos against TDP-43) were lysed and analyzed with specific antibodies in western blots to determine the endogenous TDP-43 silencing, as well as the associated expression of the studied cell and viral proteins, as similarly assayed [12].

### 4.5. RNA Extraction and RT-qPCR

RNA from HEK-293T cells was isolated using the RNeasy Mini kit (Qiagen, Hilden, Germany) following the manufacturer’s instructions. For cellular qRT-PCR (reverse transcription quantitative real-time PCR), 1000 ng of total RNA was reverse transcribed using iScript™ cDNA Synthesis Kit (Bio-Rad, Hercules, CA, USA), random primers, and anchored oligo-dT primer. The RT reaction (20 µL) was used as a template for transcript amplification; 1/10 dilutions were used in triplicate with 0.2 µM of primer and 10 µL of LightCycler 480 SYBR Green I Master in a 20 µL reaction and qPCR executed in a 96-well block on a CFX96 Touch Real-Time PCR Detection System (Bio-Rad). Absolute transcript levels for TDP-43, HDAC6, and the controls were obtained by the second derivative method. Relative transcript levels were calculated as TDP-43 or HDAC6/controls ratio and normalized to the relative expression level of the mock-transfected control. The primers used for qPCR were as follows: for TDP-43, (NM_007375.4) (Forward: 5′-GCTTCGCTACAGGAATCCAG-3′/Reverse: 5′-GATCTTTCTTGACCTGCACC-3′. Amplicon length (bp) 290); for HDAC6, (NM_006044.4) (Forward: 5′-ATGCAGCTTGCGGTTTTTGC-3′/Reverse: 5′-TGCTGAGTTCCATTACCGTGG-3′. Amplicon length (bp) 152); and for GADPH, (NM_001256799.3) as housekeeping gene (Forward: 5′-GGAAGCTCACTGGCATGGCCT-3′/Reverse: 5′-CGCCTGCTTCACCACCTTCTTG-3′. Amplicon length (bp) 119). These primers and techniques were also previously characterized [12].

### 4.6. Production of Viral Particles

Replication-deficient luciferase-HIV-1 viral particles (luciferase-reporter pseudoviruses) were obtained in packaging HEK-293T cells, as previously described [12,19,20,21,22,23,25,63,64,65,66,67] and by using the luciferase-expressing reporter virus HIV/Δnef/Δenv/luc+ (pNL4-3.Luc.R-E-provirus bearing the luciferase gene inserted into the nef ORF and not expressing env; catalog number 6070013, NIH AIDS Research and Reference Reagent Program) together with the reference Env-R5 tropic plasmid (BaL.01). Thus, HIV-1 viral particles were produced in 12-well plates by co-transfecting HEK-293T packaging cells (70% confluence) with pNL4-3.Luc.R-E- (1 µg), R5-tropic (BaL.01), and the different construct plasmids or siRNA oligos. Viral plasmids were co-transduced in HEK-293T cells using X-tremeGENE HP DNA transfection reagent (Roche Diagnostics, GmbH, Mannheim, Germany). After the addition of X-tremeGENE HP to the viral plasmids, the solution was mixed in 100 µL of DMEM medium without serum or antibiotics and was incubated for 20 min at RT prior to adding it to HEK-293T cells. The cells were cultured for 48 h to allow viral production. After this, viral particles were harvested. Viral stocks were normalized by p24-Gag content, as measured with an Enzyme Linked Immunosorbent Assay test (GenscreenTM HIV-1 Ag Assay; Bio-Rad, Marnes-la-Coquette, France). When indicated, this method for viral production was performed in packaging HEK-293T cells, also overexpressing the different plasmids for the TDP-43 constructs (Flag-wt-TDP-43 or Flag-NLS-mut-TDP-43), the control pcDNA.3.1 (-) plasmid, as well as the siRNA TDP-43 (B + C) oligos or the scrambled control oligo for the RNAi experimental condition. These plasmids or oligos were transduced by using the PEI-25k reagent as above reported. Cell-free released virions were used to infect CEM.NKR-CCR5 cells after ELISA-p24 quantification and the normalization of viral input. As a control for the efficiency of viral production (not indicated in figures), a CD4 independent viral entry and infection assay was always performed in parallel by co-transducing the pNL4-3.Luc.R-E- vector (1 µg) with the pHEF-VSV-G vector (1 µg; National Institutes of Health-AIDS Reagent Program), thereby generating non-replicative viral particles that fuse with cells in a VSV-G-dependent and CD4-independent manner, as reported [12,19,21,22,23,25,63,66].

### 4.7. Luciferase Viral Entry and Infection Assay

CEM.NKR-CCR5 cells (9 × 10^5^ cells in 24-well plates with 20 µg/mL of Polybrene) were infected with 200 ng of p24 of luciferase-reporter pseudoviruses, bearing R5-tropic BaL Env, and were produced under the different experimental conditions of this study (see previous point for viral production) in 1 mL of total volume with RPMI 1640 for 2 h (by centrifugation at 1200× *g* at 25 °C), with subsequent incubation for 4 h at 37 °C, as previously described [12,19,20,21,22,23,25,63,64,65,66,67]. Unbound viruses were then removed by washing the infected cells. After 24 h of infection, luciferase activity was measured using a luciferase assay kit (Biotium, Hayward, CA, USA) with a microplate reader (VictorTM X5; PerkinElmer, Waltham, MA, USA). Anti-CD4 neutralizing mAb L3T4 (5 μg/mL) was used as a control for the blockade of HIV-1 infection by preincubating permissive CEM.NKR-CCR5 cells with this mAb for 30 min at 37 °C before adding viral input.

### 4.8. Pseudovirus Precipitation

To study the encapsulated proteins the virions produced, a biochemical analysis of the obtained pseudoviruses was performed. For this, the supernatant from virus-producing cells was centrifuged at 3000 rpm and 4 °C for 4 min, and the pseudoviruses generated were quantified by ELISA p24 (see Luciferase viral entry and infection assay below). Later, a synchronous dose of virions (in µg/mL) was precipitated with acetone at −20 °C for 2 h (4 volumes of acetone per volume of supernatant). After this incubation, the sample was centrifuged for 15 min at 13,000 rpm (4 °C), and the resulting supernatant was decanted, leaving the remaining acetone to evaporate (10 min at RT). Then, the proteins in the pellet were treated with lysis solution for pseudoviruses (Tris-HCl 20 mM (pH 7.2), NaCl 1 M, DTT 5 mM) for 30 min (4 °C) and were sonicated for at least 30 s. The sample was centrifuged again for 15 min at 13,000 rpm (4 °C), and the viral protein pellet formed was resuspended in Laemmli solution and biochemically analyzed by western blot, as similarly reported [22].

### 4.9. Western Blotting

Protein expression was determined by SDS-PAGE and western blot in cell lysates. Plasmids or siRNA oligos were transfected into HEK-293T cells using PEI-25k (or by nucleofection). The plasmid/oligos-PEI-25k mixture was gently vortexed, incubated for 20–30 min at RT, and then added to cells in culture. Briefly, 48 h after transfection for both over-expression constructs and siRNA oligos, cells were lysated in lysis buffer (1% Triton-X100, 50 mM Tris-HCl pH 7.5, 150 mM NaCl, 0.5% sodium deoxycholate, and protease inhibitor (Roche Diagnostics)) for 30 min and sonicated for 30 s at 4 °C. The effects of the different inhibitors were similarly assayed in HEK-293T cells. For this purpose, 24 h post-transfection cells were treated for 5 h at 37 °C with any of the following inhibitors: MG132, 20 µM dissolved in dimethyl sulfoxide (DMSO) to inhibit the proteasome; 3-MA, 5 mM in PBS to inhibit autophagosome formation and subsequent autophagic degradation, monitored by detecting p62/SQSTM1 protein. Equivalent amounts of protein (30–40 μg), determined using the bicinchoninic acid (BCA) method (Millipore Corporation, Billerica, MA, USA) were resuspended and treated by Laemmli buffer and then were separated in 10% SDS-PAGE and electroblotted onto 0.45 μm polyvinylidene difluoride membranes (PVDF; Millipore) using Trans-blot Turbo (Bio-Rad, Hercules, CA, USA). Membranes were blocked by 5% non-fat dry milk in TBST (100 mM Tris, 0.9% NaCl, pH 7.5, 0.1% Tween 200) for 30 min and then incubated with specific antibodies. Proteins were detected by luminescence using the ECL System (Bio-Rad) and were analyzed using a ChemiDoc MP device and Image LabTM Software, Version 5.2 (Bio-Rad), as reported [12,21,22].

### 4.10. Statistical Analysis

Statistical analyses were performed using GraphPad Prism, version 6.0b (GraphPad Software), and using Student’s *t* test as indicated in figure legends.

## Figures and Tables

**Figure 1 ijms-24-07658-f001:**
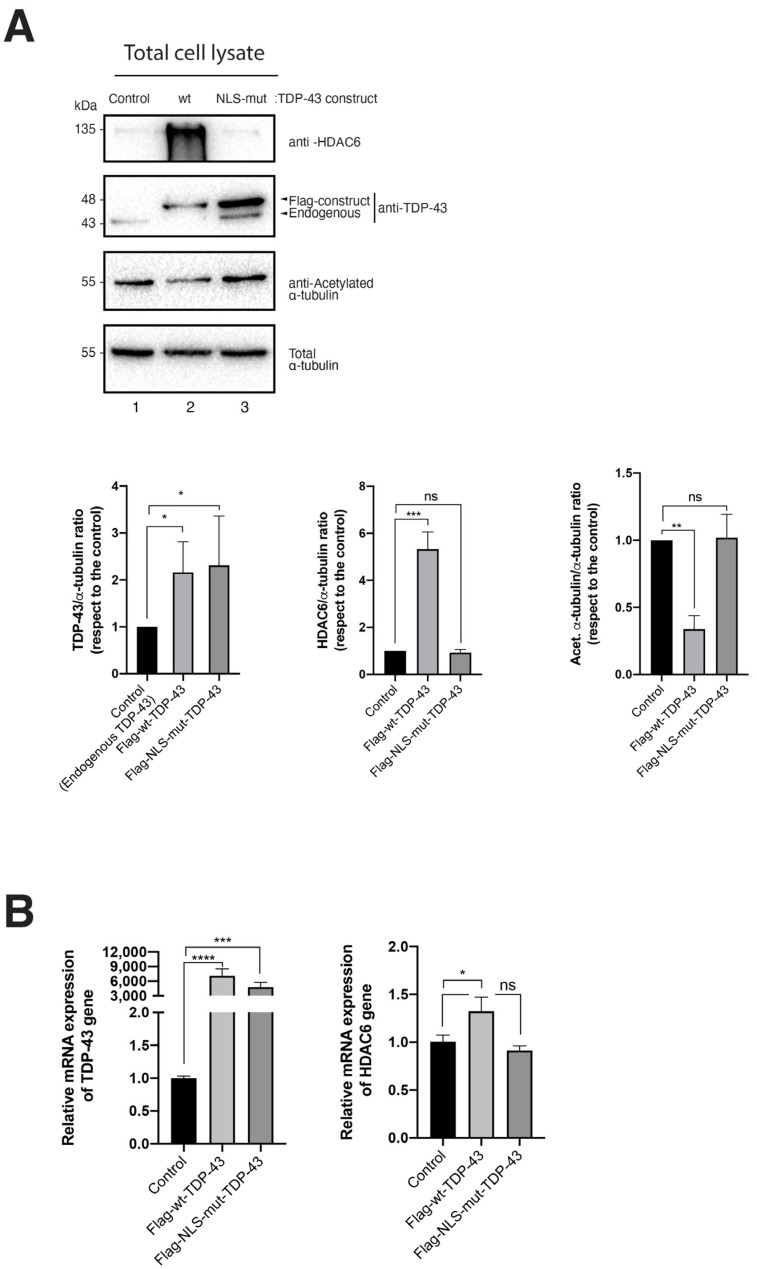
Overexpression of TDP-43 increases HDAC6 mRNA and protein levels. (**A**) Quantitative biochemical analysis of whole cell lysate of HEK-293T cells overexpressing Flag-wt-TDP-43 (line 2 condition) or the Flag-NLS-mut-TDP-43 construct (line 3 condition) and their effect on endogenous HDAC6 and MTs α-tubulin acetylation, with respect to control condition (line 1). Acetylated α-tubulin levels are a read-out for MT stabilization that decrease after increasing tubulin-deacetylase HDAC6. Total α-tubulin for protein load control is shown, under each experimental condition. A representative experiment of three is shown (see replicate data in Appendix A section). *Bottom*: Histograms quantify the intensities of western blot bands from the left panel experiment, representing the amounts of TDP-43 constructs (Flag-wt-TDP-43 and Flag-NLS-mut-TDP-43) compared to endogenous TDP-43 (control), HDAC6, and acetylated α-tubulin. Data are normalized by total α-tubulin load control per each experimental condition. Data are mean ± standard error of the mean (S.E.M.) of three independent experiments (data associated with the quantification of replicates in Appendix A section). When indicated, *** *p* < 0.001 and ** *p* < 0.05 are *p* values for Student’s *t* test. ns stands for non-significant. (**B**) Relative mRNA quantification by RT-qPCR (4 repeats) of TDP-43 and HDAC6 genes are represented in histograms under the overexpression of TDP-43 constructs. When indicated, * *p* < 0.05, ** *p* < 0.01, *** *p* < 0.001, and **** *p* < 0.0001 are *p* values for Student’s *t* test; ns, not significant.

**Figure 2 ijms-24-07658-f002:**
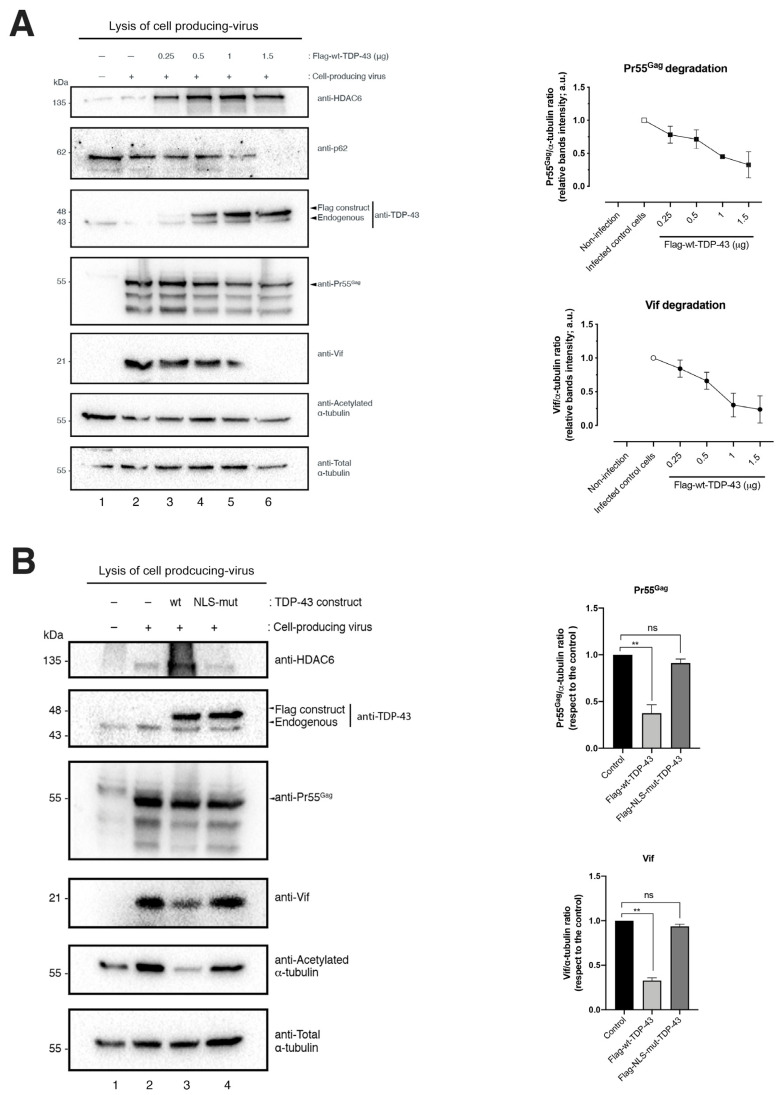
TDP-43 overexpression stabilizes HDAC6 and degrades HIV-1 Pr55^Gag^ and Vif viral proteins. (**A**) Biochemical western blot analysis of Flag-wt-TDP-43 overexpression in cell lysates of virus-producing HEK-293T cells (Env-BaL + viral backbone pNL4-3.luc.E-R- (see Materials and Methods section)) transduced with the Flag-wt-TDP-43 construct in an increasing plasmid dose (0, 0.25, 0.50, 1, and 1.5 μg). The level of expression of the Pr55^Gag^ and Vif viral proteins were also analyzed together with stabilized HDAC6 and its acetylated α-tubulin substrate. Total α-tubulin for protein load control is shown under each experimental condition. A representative experiment of three is shown (see replicate data in Appendix A section). The quantification of the Pr55^Gag^ and Vif clearance, with respect to the load control protein, total α-tubulin, was analyzed in histograms on the right. Data are mean ± S.E.M. of three independent experiments (data associated with the quantification of replicates in Appendix A section). (**B**) Biochemical western blot analysis and quantification (right histograms, with respect to the load control protein total α-tubulin) of the wt-TDP-43- and NLS-TDP-43 mutant-mediated effect on HIV-1 Pr55^Gag^ and Vif protein clearance. Total α-tubulin for protein load control is shown under each experimental condition. Histogram-associated data for HIV-1 Pr55^Gag^ and Vif degradation are the mean ± S.E.M. of three independent experiments (data associated with the quantification of replicates in Appendix A section). The quantification of the Pr55^Gag^ and Vif clearance, with respect to the load control protein total α-tubulin, was analyzed in histograms on the right. Data are mean ± S.E.M. of three independent experiments (data associated with the quantification of replicates in Appendix A section). When indicated, ** *p* < 0.01 is the *p* value for Student’s *t* test; ns, not significant.

**Figure 3 ijms-24-07658-f003:**
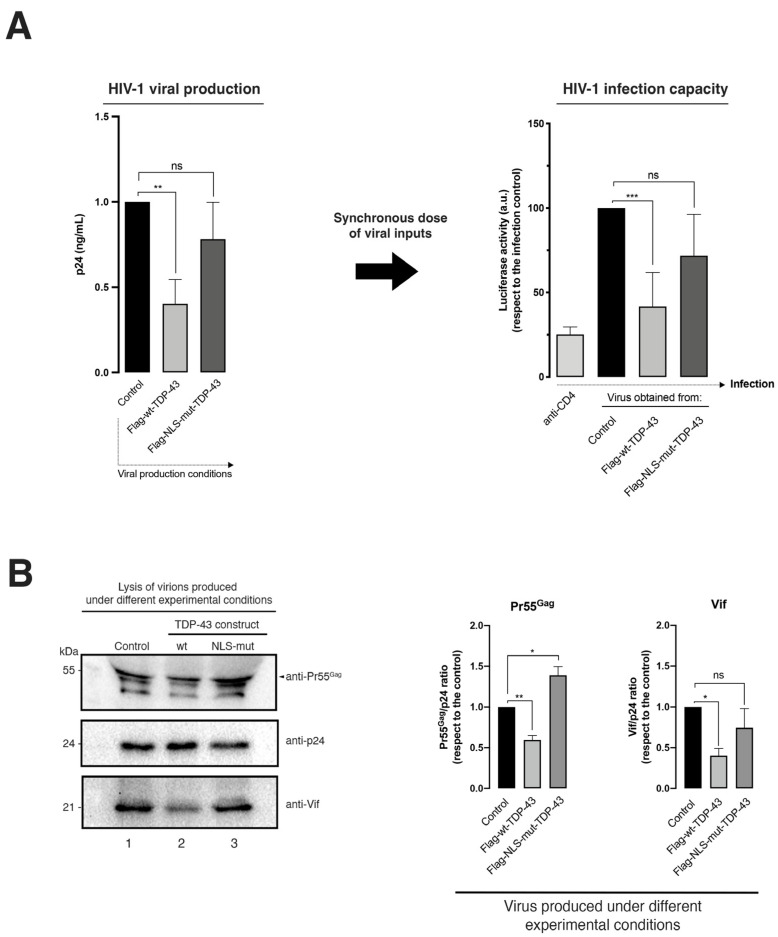
TDP-43 controls HIV-1 viral production and infection capacity by regulating the amount of Pr55^Gag^ and Vif viral proteins incorporated into viral particles. (**A**) *Left*: ELISA-HIV-1 p24 protein quantification of produced viral particles, under different TDP-43-experimental conditions. The ELISA-p24 technique was standardized with respect to the viral production control, which corresponded to non-treated cell experimental conditions (control; endogenous TDP-43). Data are the mean ± S.E.M. of three independent experiments. *Right*: HIV-1 infection was then carried out with synchronous doses of luciferase-pseudoviruses produced under the TDP-43 experimental conditions of panel (**A**), in permissive CEM.NKR-CCR5 CD4+ T cells. CD4-treated cells showed the neutralization of HIV-1 infection by an anti-CD4 antibody (Ab). Data are the mean ± S.E.M. of three independent experiments. (**B**) Biochemical western blot analysis of the content of Pr55^Gag^, p24, and Vif proteins of the pseudoviruses used to infect permissive cells of the experiment of panel (**A**); experimental condition of Figure 2B. A representative experiment of three is shown (see replicate data in Appendix A section). The quantification of the viral content in Pr55^Gag^ and Vif proteins, normalized by the p24 protein as a control for synchronous viral input, is shown in the right histograms under each experimental condition. Data associated with the quantification of replicates are shown in Appendix A section. In (**A**,**B**), data are the mean ± S.E.M. of three experiments independent in triplicate. When indicated, * *p* < 0.05, ** *p* < 0.01, and *** *p* < 0.001 are the *p* value for Student’s *t* test; ns, not significant; a.u., arbitrary light units.

**Figure 4 ijms-24-07658-f004:**
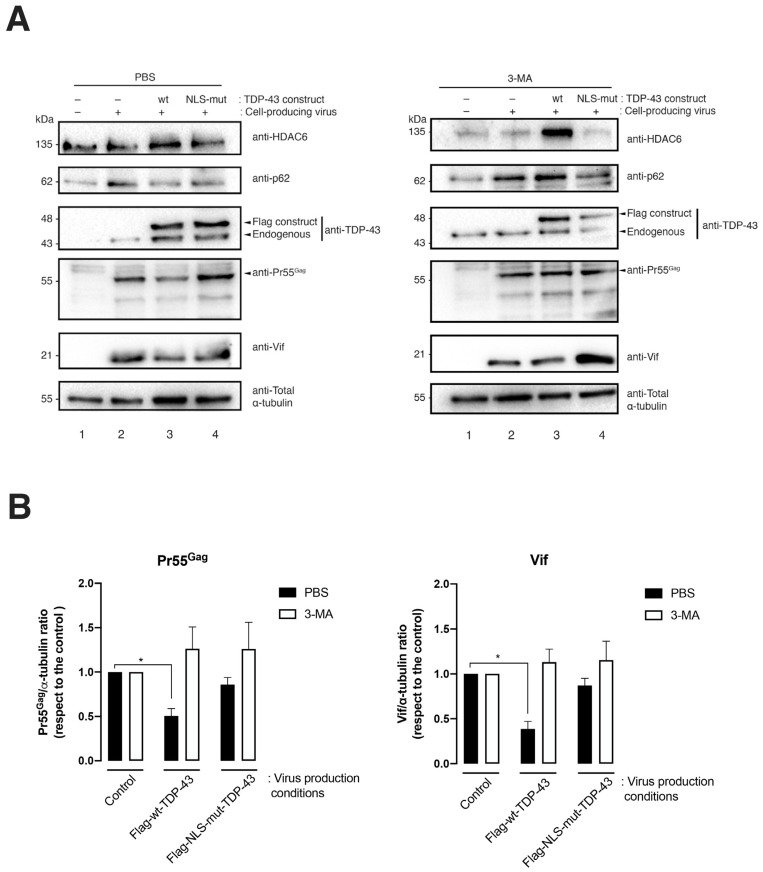
TDP-43 modulates the level of expression of the HIV-1 Pr55^Gag^ and Vif proteins by promoting HDAC6 protein expression and autophagy degradation. (**A**) The biochemical western blot analysis of Pr55^Gag^ and Vif degradation together with HDAC6, p62, and acetylated α-tubulin stabilization in virus-producing HEK-293T cells transfected with Flag-wt-TDP-43 and Flag-NLS-mut-TDP-43 and treated with the 3-MA inhibitor (5 mM) or vehicle control (PBS). Total α-tubulin for protein load control is shown under each experimental condition. A representative experiment of three is shown (see replicate data in Appendix A section). (**B**) Histograms show the quantification of western blot bands for the Pr55^Gag^/or Vif/α-tubulin ratio under inhibitor (3-MA) or control (PBS) experimental conditions. Data associated with the quantification of replicates are shown in Appendix A section. Data are represented as the mean ± S.E.M. of three independent experiments. When indicated, * *p* < 0.05 is the *p* value for Student’s *t* test.

**Figure 5 ijms-24-07658-f005:**
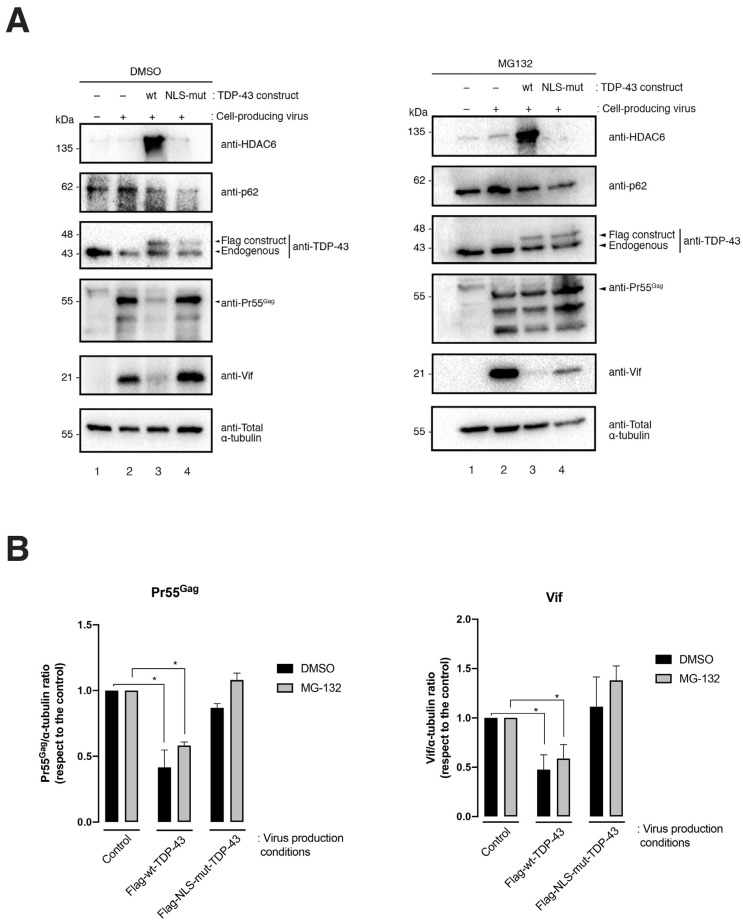
TDP-43 mediated clearance of the HIV-1 Pr55^Gag^ and Vif viral proteins does not occur by the proteasome pathway. (**A**) The biochemical western blot analysis of HIV-1 Pr55^Gag^ and Vif degradation together with HDAC6 and p62 protein expression levels in virus-producing HEK-293T cells transfected with Flag-wt-TDP-43 or Flag-NLS-mut-TDP-43, treated or not with the proteasome chemical inhibitors MG132 (20 µM). DMSO is the vehicle control. Total α-tubulin for protein load control is shown under each experimental condition. A representative experiment of three is shown (see replicate data in Appendix A section). (**B**) Histograms show the quantification of western blot bands for the Pr55^Gag^/or Vif/α-tubulin ratio under inhibitor (MG132) or control (DMSO) experimental conditions. Data associated with the quantification of replicates are shown in Appendix A section. Data are represented as the mean ± S.E.M. of three independent experiments. When indicated, * *p* < 0.05 is the *p* value for Student’s *t* test.

**Figure 6 ijms-24-07658-f006:**
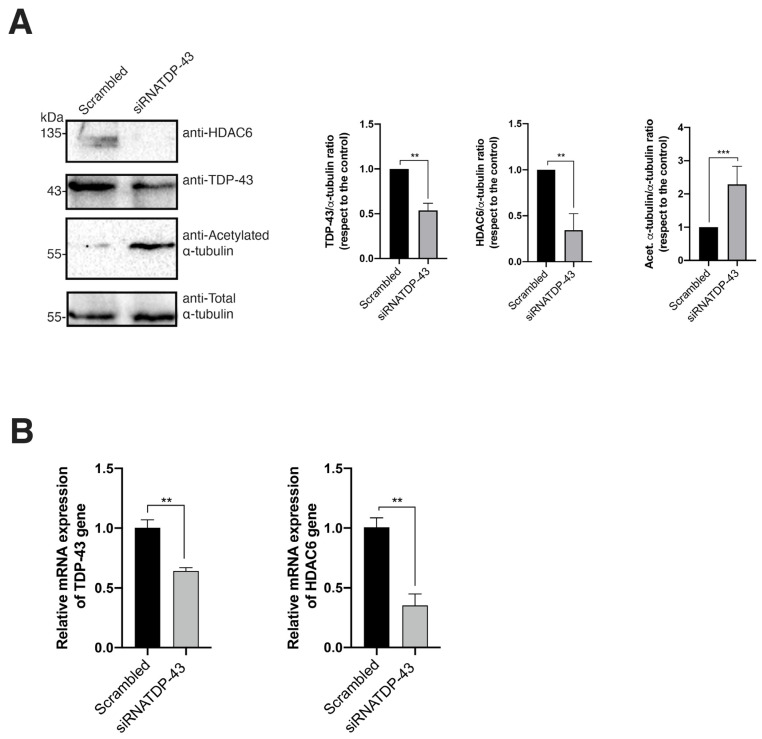
Specific TDP-43 mRNA silencing diminishes mRNA and protein levels of HDAC6. (**A**) The biochemical western blot analysis of endogenous TDP-43 and HDAC6 protein expression levels in HEK-293T cells transfected with a mix of siRNA oligonucleotides (siRNA-TDP-43 (B + C)) against TDP-43 (siRNATDP-43 experimental condition) or the control, which was scrambled-treated cells. The HDAC6 subtract acetylated α-tubulin in MTs is shown. Total α-tubulin for protein load control is shown under each experimental condition. Data are represented as the mean ± S.E.M. of three independent experiments (see replicate data in Appendix A section). Histograms show the quantification of western blot bands for the TDP-43/, HDAC6/, and acetylated α-tubulin/α-tubulin ratio under this experimental condition. Data associated with the quantification of replicates are shown in Appendix A section). (**B**) The relative quantification of TDP-43 and HDAC6 mRNA by RT-qPCR (3 replicates) is represented in histograms in siRNA-TDP-43-treated HEK-293T cells or the control, which was scrambled-treated cells. In (**A**,**B**), when indicated, ** *p* < 0.001 and *** *p* < 0.001 are the *p* values for Student’s *t* test.

**Figure 7 ijms-24-07658-f007:**
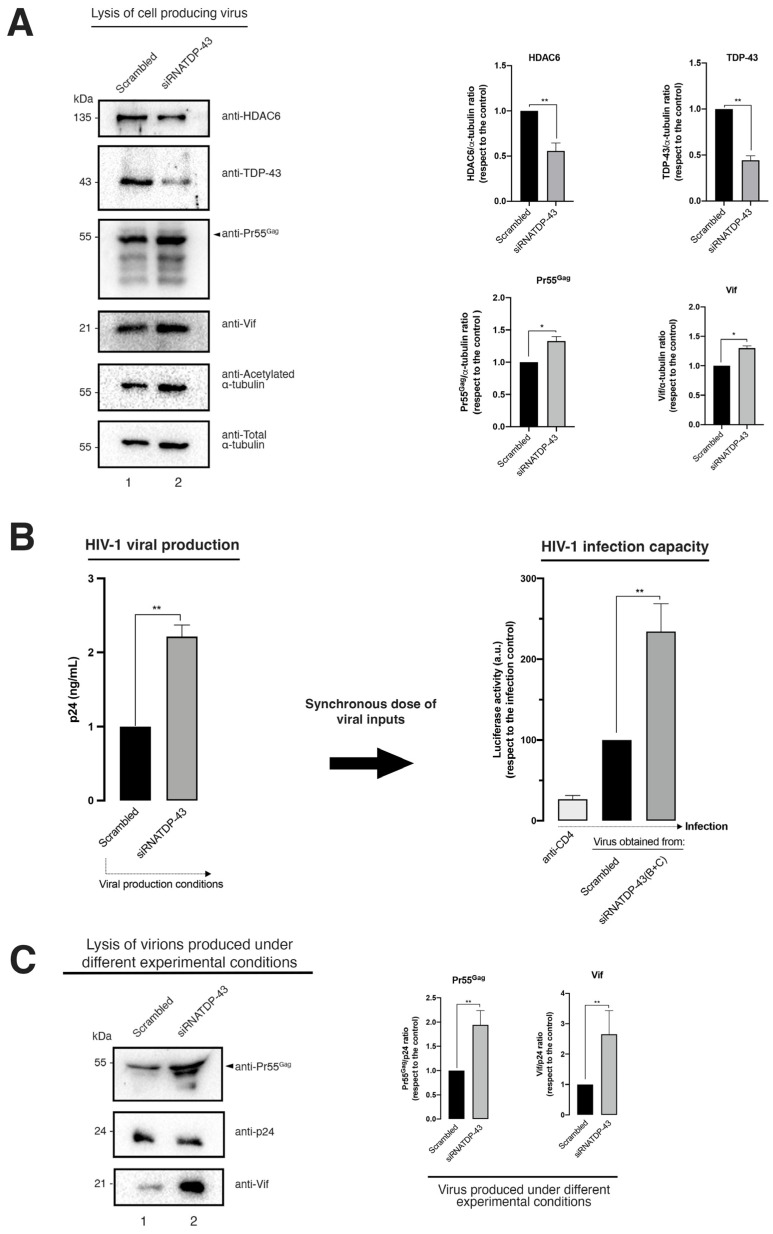
The specific silencing of TDP-43 in virus-producing cells decreases the protein level of HDAC6, stabilizing acetylated α-tubulin and HIV-1 Pr55^Gag^ and Vif viral proteins, thereby enhancing the production of viral particles and their infectivity. (**A**) The biochemical western blot analysis of TDP-43, HDAC6, acetylated α-tubulin, HIV-1 Pr55^Gag^, and Vif proteins in virus-producing HEK-293T cells treated with a pull of siRNA oligonucleotides specific against TDP-43 (siRNATDP-43) or in the control (scrambled-treated cells). Total α-tubulin for protein load control is shown under each experimental condition. Data are represented as the mean ± S.E.M. of three independent experiments (see replicate data in Appendix A section). *Right*: Histograms show the quantification of western blot bands for the TDP-43/, HDAC6/, Pr55^Gag^/, and Vif/α-tubulin ratios under this experimental condition. Data associated with the quantification of replicates are shown in Appendix A section. (**B**) *Left*: ELISA-HIV-1 p24 protein quantification of produced viral particles and siRNA-TDP-43- or scrambled-treated virus-producing HEK-293T cells. The ELISA-p24 technique was standardized with respect to the viral production control, which corresponded to the experimental condition of the scrambled-treated cells. Data are mean ± S.E.M. of three independent experiments. *Right*: HIV-1 infection was then carried out with synchronous doses of luciferase-pseudoviruses produced under the experimental conditions of panels (**A**,**B**) in the permissive CEM.NKR-CCR5 CD4+ T cells. CD4-treated cells showed the neutralization of HIV-1 infection by an anti-CD4 Ab. Data are mean ± S.E.M. of three independent experiments. (**C**) The biochemical western blot analysis of the content of Pr55^Gag^, p24, and Vif proteins of the pseudoviruses used to infect permissive cells of the experiment of panel (**B**). A representative experiment of three is shown (see replicate data in Appendix A section). The quantification of the viral content in Pr55^Gag^ and Vif proteins, normalized by the p24 protein as a control for synchronous viral input, is shown in the right histograms, under each experimental condition. Data associated with the quantification of replicates are shown in Appendix A section. In (**A**–**C**), when indicated, * *p* < 0.05 and ** *p* < 0.01 are the *p* values for Student’s *t* test; a.u., arbitrary light units.

**Figure 8 ijms-24-07658-f008:**
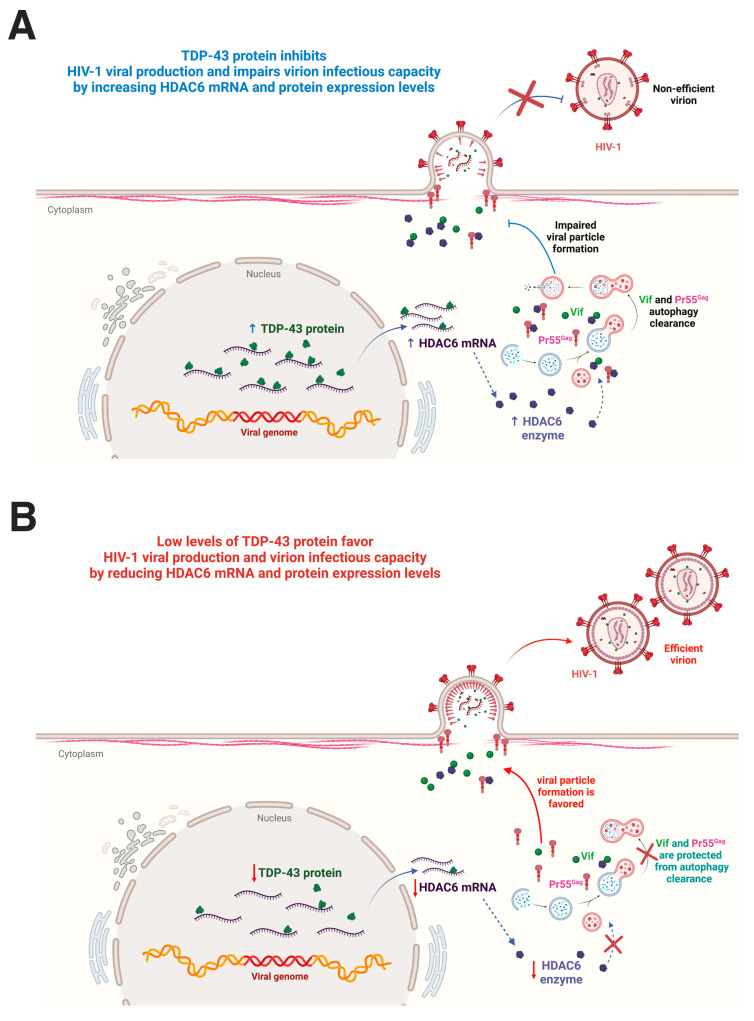
Scheme summarizing the TDP-43 control of HIV-1 viral particle production and virus infectiveness. (**A**) The overexpression of functional TDP-43 leads to an increase of HDAC6 mRNA and protein levels, which promote the autophagic clearance of HIV-1 Pr55^Gag^ and Vif proteins, thereby inhibiting viral particle production and virus infectivity. These viral particles incorporate a reduced amount of HIV-1 Pr55^Gag^ and Vif proteins. (**B**) Endogenous TDP-43 silencing reduces HDAC6 expression (i.e., mRNA and protein), stabilizes HIV-1 Vif and Pr55^Gag^ proteins, and favors their incorporation into virions, thereby promoting HIV-1 viral particle production and infectious capacity. Thus, there is a direct relationship between the content of HIV-1 Vif and Pr55^Gag^ proteins in virions and their infection capacity. Therefore, TDP-43 (i.e., the TDP-43/HDAC6 axis) could be considered a key factor to control HIV-1 viral production and virus infectiveness. Designs and templates were created with BioRender.

## Data Availability

All relevant datasets are contained within the manuscript. Please, see in “**Appendix A”** the Appendix A containing the replicates of all western blot biochemical analysis of data presented in this work.

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
