# Peer review of "TDP-43 Controls HIV-1 Viral Production and Virus Infectiveness"

_ijms, 2023, doi:10.3390/ijms24087658_

Round 1

Reviewer 1 Report

In this manuscript, the author described “TDP-43 controls HIV-1 viral production and virus infectiveness”. TDP-43, stabilizing HDAC6 mRNA and protein levels, negatively affects HIV-1 viral production and virus infectiveness by targeting Pr55Gag and Vif viral proteins. Additionally, the author also showed degradative action of TDP-43 on HIV-1 Pr55Gag and Vif proteins is mediated by autophagy. These degradative events impair the production of viral particles and reduce the infectious capacity of nascent virions. This is a well-written manuscript, the author made a great effort, Overall, the manuscript is improved but I have some minor comments for the study that authors can consider.

comments

1.     In figure 3A, please explain why there is some value in uninfected samples.

2.     In Figure 4, please write at what concentration 3-MA was used.

3.     In figure 5, please write at what concentration MG-132 was used.

4.     In figure 6, Author did not show the cell viability assay for siRNA.

This is a well-written manuscript, but still minor editing is required in introduction and discussion part.

Author Response

Reply to the Reviewer #1:

Comments and Suggestions for Authors:

In this manuscript, the author described “TDP-43 controls HIV-1 viral production and virus infectiveness”. TDP-43, stabilizing HDAC6 mRNA and protein levels, negatively affects HIV-1 viral production and virus infectiveness by targeting Pr55Gag and Vif viral proteins. Additionally, the author also showed degradative action of TDP-43 on HIV-1 Pr55Gag and Vif proteins is mediated by autophagy. These degradative events impair the production of viral particles and reduce the infectious capacity of nascent virions. This is a well-written manuscript, the author made a great effort. Overall, the manuscript is improved but I have some minor comments for the study that authors can consider.

We thank Reviewer#1 for this summary of our work and the kind comments on our work also highlighting the role of the TDP-43/HDAC6 axis on the control of HIV-1 replication and virion infection capacity.

Comments on the Quality of English Language:

This is a well-written manuscript, but still minor editing is required in introduction and discussion part.

As suggested by the Reviewer #1, we have English corrected the Introduction and the Discussion section (red-yellow-text marked in these sections of the revised Manuscript (ID: ijms-2356206)) by using “The American Journal Experts (AJE) service” by the “Research Square Company (Durham, North Carolina, USA)“ (https://www.aje.com/?_ga=2.118227210.280156449.1681374076-1988308735.1679566133).

Comments:

  1. In figure 3A, please explain why there is some value in uninfected samples.

The values found in p24 protein ELISA quantification (Figure 3A, left histograms) and luciferase activity assay (Figure 3A, right histograms) correspond to the background measurements of the techniques assayed by using control non-virus producing and non-infected cells, respectively. These values are therefore non-specific reads of the techniques used with negative control cell lysates. Noteworthy, in the case of HIV-1 infection, productive infection is inhibited by using a neutralizing anti-CD4 antibody. Therefore, and to facilitate the comprehension of the data presented in this Figure 3A, we have removed from the revised Figure 3A the non-specific background values obtained under negative control conditions.

  1. In Figure 4, please write at what concentration 3-MA was used.

&

  1. In figure 5, please write at what concentration MG-132 was used.

 We would like to thank Reviewer #1 for these two associated remarks. We have specified the working concentrations used for the 3-MA inhibitor, in the legend to the Figure 4, and for the MG132 inhibitor in the legend to the Figure 5. Likewise, we have indicated this information in the “4.9 western blotting (Materials and methods)” section (lines 832-837), as follows:

“The effects of the different chemical inhibitors used were similarly assayed in HEK-293T cells. For this purpose, 24h post-transfection cells were treated for 5 h at 37oC with the following inhibitors: 20 mM of MG132, dissolved in dimethyl sulfoxide (DMSO), to inhibit the proteasome; 5 mM of 3-MA in PBS to inhibit aggresome formation and subsequent autophagic degradation, monitored by detecting p62/SQSTM1 protein.

  1. In figure 6, Author did not show the cell viability assay for siRNA.

We would like thank Reviewer #1 for this comment. Below, we have included some representative data regarding assays for viability (i.e., measurement of cellular propidium iodide uptake by flow cytometry and counting trypan blue-stained cells under a light microscope) of virus producing cells treated with scrambled or siRNA-TDP-43 oligos. These are rutinary controls performed in our Lab and we have not included these data as part of the associated Figure 6. However, we have indicated these procedures in the Materials and methods section (lines 725-729), and referred to a previous reported work of our group for the propidium iodide uptake assay (Barroso-González J, et al. The lupane-type triterpene 30-oxo-calenduladiol is a CCR5 antagonist with anti-HIV-1 and anti-chemotactic activities. J Biol Chem. 2009 Jun 12;284(24):16609-16620. doi: 10.1074/jbc.M109.005835. Epub 2009 Apr 22. PMID: 19386595; PMCID: PMC2713542). Cells are viable under these experimental conditions.

Please, see representative cell viability data in the associated Figure in the attached PDF file, including the entire point-by-point reply.

Figure.  Specific siRNA-TDP-43 or scrambled oligos are not toxic for virus producing HEK-293T cells.

(A) Flow cytometry-based analysis of propidium iodide uptake in control, scrambled-treated HEK-293T cells (top left panel) and siRNA-TDP-43-transfected HEK-293T cells (24 h post-transfection). Quantification of propidium iodide uptake (FL3) by cells is indicated in UR (upper right) regions of plots, per each experimental condition. The constitutive endogenous CXCR4 cell-surface expression is detected by using a specific PE-conjugated mAb (FL2), as a control for a marker that is not affected by TDP-43 knockdown, under any experimental condition as reported (Cabrera-Rodríguez R, et al. Transactive Response DNA-Binding Protein (TARDBP/TDP-43) Regulates Cell Permissivity to HIV-1 Infection by Acting on HDAC6. Int J Mol Sci. 2022 May 31;23(11):6180. doi: 10.3390/ijms23116180). Basal cell fluorescence intensity for CXCR4 receptor labelling was determined by staining cells with a PE-conjugated IgG isotype control. Flow cytometry was performed by using a BD-Accuri™ C6 Plus Flow Cytometer (BD Biosciences, San Jose, CA, USA). A representative experiment is shown. (B) In this test for cell trypan blue stain, virus producing, non-treated, scrambled- or siRNA-TDP-43-treated HEK-293T cells are mixed in suspension with trypan blue dye (1 part of 0.4% trypan blue and 1 part cell suspension), during 3 min at room temperature, then visualizing cells under a light microscope to determine whether cells take up or exclude the dye. 100 cells were counted per each experimental condition at 0, 24h and 48h of treatment. Hence, viable cells will have a clear cytoplasm whereas a nonviable cell will have a blue cytoplasm, as reported (Strober W. Trypan Blue Exclusion Test of Cell Viability. Curr Protoc Immunol. 2015 Nov 2;111:A3.B.1-A3.B.3. doi: 10.1002/0471142735.ima03bs111. PMID: 26529666; PMCID: PMC6716531). Data are mean ± S.E.M. of three independent experiments.

Noteworthy, we would like also to outstand that the siRNA oligos used in our research work were assayed as previously reported by our team (Cabrera-Rodríguez R, et al. Transactive Response DNA-Binding Protein (TARDBP/TDP-43) Regulates Cell Permissivity to HIV-1 Infection by Acting on HDAC6. Int J Mol Sci. 2022 May 31;23(11):6180. doi: 10.3390/ijms23116180. PMID: 35682862; PMCID: PMC9181786) and others (Fiesel FC, et al. Knockdown of transactive response DNA-binding protein (TDP-43) downregulates histone deacetylase 6. EMBO J. 2010 Jan 6;29(1):209-21. doi: 10.1038/emboj.2009.324. Epub 2009 Nov 12. PMID: 19910924; PMCID: PMC2808372). Cells were quantified after 24h and 48h of siRNA treatment and assayed for cell viability, as above indicated. Likewise, TDP-43-silenced cells were assayed by RT-qPCR, where we sought good expression levels of housekeeping genes (i.e., see also mRNA sequencing data in our previous work by Cabrera-Rodríguez R, et al. Transactive Response DNA-Binding Protein (TARDBP/TDP-43) Regulates Cell Permissivity to HIV-1 Infection by Acting on HDAC6. Int J Mol Sci. 2022 May 31;23(11):6180. doi: 10.3390/ijms23116180. PMID: 35682862; PMCID: PMC9181786), MTs post-transductional modification by alpha-tubulin acetylation (after TDP-43 knockdown due to a decrease of the HDAC6 enzyme) and an increase of HIV-1 replication with concomitant stabilization of Pr55Gag and Vif viral proteins. If cell viability were therefore compromised, under these experimental conditions, we wouldn’t observe cytoskeletal stabilization, housekeeping genes expression and infectious HIV-1 viral production, nor refringent cells under microscope after trypan-blue treatment (for cell quantification) neither low cellular uptake of propidium iodide by flow cytometry.

We would like to thank Reviewer #1 for all these comments that have improved our work.

Reviewer 2 Report

In this manuscript, Cabrera-Rodriguez et al studied how TDP-43 overexpression and silencing affects the late stages of HIV-1 life cycle. They found that overexpression of TDP-43 leads to increased level of HDAC6 mRNA and protein, and a decreased level of HIV Gag and Vif proteins. In contrast, TDP-43 knockdown reduces the level of HDAC6 expression, and elevates the level of HIV Gag and Vif. The changed level of Gag and Vif proteins lead to a corresponding diminished or enhanced viral infectivity. The authors suggested TDP-43/HDAC6 could be considered as new targets for anti-HIV-1 strategies. 

My specific comments are:

1. In this study, the effect of TDP-43 overexpression and silencing is only tested in the HEK-293T cells. A 2014 Plos One paper (Nehls et al) showed that TDP-43 knockdown has no influence on p24 release in macrophages. Due to the potential cell line dependency, it is better to show that the anti-viral activity of TDP-43 remains in at least one other cell line. 

2. In section 2.2, it sounds certain that the decreased level of Gag and Vif proteins are due to the HDAC6 stabilization. However, this was not directly tested in this manuscript. I suggest rephrasing some of the sentences (for example, line 174, "This wt-TDP-43 action on HDAC6 leads to a clearance..., this sounds like it was directly tested). 

3. Related to comment 2, could the authors comment on whether there could be any other explanations for the decreased level of Gag and Vif proteins? For example, could the viral transcription be impaired when overexpressing TDP-43?

4. Figure 3B and 7C, what does this decreased and increased level of full-length Gag imply? Since the level of Gag is normalized to the p24, does this imply that the proteolytic cleavage is more efficient in the case of less infectious virions, and less efficient in the case of more infectious virions?  

Author Response

Reply to the Reviewer #2:

Comments and Suggestions for Authors

In this manuscript, Cabrera-Rodriguez et al studied how TDP-43 overexpression and silencing affects the late stages of HIV-1 life cycle. They found that overexpression of TDP-43 leads to increased level of HDAC6 mRNA and protein, and a decreased level of HIV Gag and Vif proteins. In contrast, TDP-43 knockdown reduces the level of HDAC6 expression, and elevates the level of HIV Gag and Vif. The changed level of Gag and Vif proteins lead to a corresponding diminished or enhanced viral infectivity. The authors suggested TDP-43/HDAC6 could be considered as new targets for anti-HIV-1 strategies.

We thank Reviewer#2 for this summary of our work and the kind comments on our work also highlighting the role of the TDP-43/HDAC6 axis on HIV-1 infection.

My specific comments are:

  1. In this study, the effect of TDP-43 overexpression and silencing is only tested in the HEK-293T cells. A 2014 Plos One paper (Nehls et al) showed that TDP-43 knockdown has no influence on p24 release in macrophages. Due to the potential cell line dependency, it is better to show that the anti-viral activity of TDP-43 remains in at least one other cell line.

We would like to thank the Reviewer for arising this comment.

In a previous work, we have assayed the effects of specific knockdown of TDP-43 by siRNA on HIV-1 in different cell models for studying viral infection (Cabrera-Rodríguez R, et al. Transactive Response DNA-Binding Protein (TARDBP/TDP-43) Regulates Cell Permissivity to HIV-1 Infection by Acting on HDAC6. Int J Mol Sci. 2022 May 31;23(11):6180. doi: 10.3390/ijms23116180. PMID: 35682862; PMCID: PMC9181786). HEK-293T is a well-stablished model used to characterize the late steps of the HIV-1 viral cycle, allowing the quantification of viral particle production. Moreover, we have characterized the silencing of endogenous TDP-43 by quantifying mRNA and protein levels of TDP-43, as well as the associated HDAC6 transcript and enzyme, together with functional process involved in HIV-1 infection, such as microtubules stabilization and autophagy function.

In the case of Nehls et al. work (Nehls J, et al. HIV-1 replication in human immune cells is independent of TAR DNA binding protein 43 (TDP-43) expression. PLoS One. 2014 Aug 15;9(8):e105478. doi: 10.1371/journal.pone.0105478. PMID: 25127017; PMCID: PMC4134290), the authors introduce HIV-1 material in macrophages by using a CD4/coreceptor-independent route by using pseudotyped virions bearing the VSV-G envelope protein. This envelope drives viral entry by an endocytic pathway that overcomes the necessity of HIV-1 Env to signal to productively infect cells, and therefore to stabilize and reorganize cytoskeleton and regulate autophagy (Valenzuela-Fernández A, et al. Histone deacetylase 6 regulates human immunodeficiency virus type 1 infection. Mol Biol Cell. 2005 Nov;16(11):5445-54. doi: 10.1091/mbc.e05-04-0354. Epub 2005 Sep 7. PMID: 16148047; PMCID: PMC1266439 ; Barrero-Villar M, et al. Moesin is required for HIV-1-induced CD4-CXCR4 interaction, F-actin redistribution, membrane fusion and viral infection in lymphocytes. J Cell Sci. 2009 Jan 1;122(Pt 1):103-13. doi: 10.1242/jcs.035873. Epub 2008 Dec 9. PMID: 19066282 ; Santos G, Valenzuela-Fernández A, Torres NV. Quantitative analysis of the processes and signaling events involved in early HIV-1 infection of T cells. PLoS One. 2014 Aug 8;9(8):e103845. doi: 10.1371/journal.pone.0103845. Erratum in: PLoS One. 2015;10(2):e0117867. PMID: 25105875; PMCID: PMC4126662 ;   Pérez-Yanes S, et al. The Characteristics of the HIV-1 Env Glycoprotein Are Linked With Viral Pathogenesis. Front Microbiol. 2022 Mar 24;13:763039. doi: 10.3389/fmicb.2022.763039. PMID: 35401460; PMCID: PMC8988142 ; Cabrera-Rodríguez R et al. HIV-1 envelope glycoproteins isolated from Viremic Non-Progressor individuals are fully functional and cytopathic. Sci Rep. 2019 Apr 3;9(1):5544. doi: 10.1038/s41598-019-42075-3. PMID: 30944395; PMCID: PMC6447548 ; Cabrera-Rodríguez R, et al. The Interplay of HIV and Autophagy in Early Infection. Front Microbiol. 2021 Apr 28;12:661446. doi: 10.3389/fmicb.2021.661446. PMID: 33995324; PMCID: PMC8113651 ; Valenzuela-Fernández A, et al. Contribution of the HIV-1 Envelope Glycoprotein to AIDS Pathogenesis and Clinical Progression. Biomedicines. 2022 Sep 2;10(9):2172. doi: 10.3390/biomedicines10092172. PMID: 36140273; PMCID: PMC9495913). Hence, Nehls and coworkers introduce massive HIV-1 backbone material by using a VSV-G-mediated cell entry, avoiding the Env-mediated infection process. This fact could assure the LTR-mediated activation that leads to the expression of high amounts of the p24 viral protein, which is produced at least during 24h before macrophage treatment with siRNA-TDP-43 oligos. The amount of p24 released could be conditioned by this window period of time. Likewise, before introducing HIV-1/VSV-G material, macrophages were treated by the Vpx viral protein. It means that the authors mix auxiliary viral factors of HIV-2 (i.e., Vpx which is coded almost exclusively by members of the SIVSM/HIV-2 lineage of primate lentiviruses) and HIV-1 (i.e., Vif) that promote the degradation of cell-antiviral restriction factors to study the effect of TDP-43 silencing in a process of infection, and during 24h before treating cells with siRNA oligos. Moreover, the authors indicate that they measure GFP expression 96h post-transfection, as well as TDP-43 western blot biochemical analysis, but they do not clarify the time used for measuring p24 release (see the legend to the Figure 6 and Material and methods section). It is important to note that HIV infected macrophages contain large intracellular pools of infectious virus that can be released from intact cells in an exosome-dependent pathway (e.g., Kramer B, et al. HIV interaction with endosomes in macrophages and dendritic cells. Blood Cells Mol Dis. 2005 Sep-Oct;35(2):136-42. doi: 10.1016/j.bcmd.2005.06.006. PMID: 16087369) or from late endosomes where the majority of infectious HIV viral particles assemble and egress (e.g., Pelchen-Matthews A, et al. Infectious HIV-1 assembles in late endosomes in primary macrophages. J Cell Biol. 2003 Aug 4;162(3):443-55. doi: 10.1083/jcb.200304008. Epub 2003 Jul 28. PMID: 12885763; PMCID: PMC2172706). Therefore, it is plausible that under the experiential procedure used by Nehls and coworkers, the p24 measured could be released from these cellular compartments and away from an antiviral TDP-43 action. Moreover, in this study, the authors did not present data of the action of Vpx on SAMHD1 or TDP-43. Moreover, the p24 measurements are not rated to the number of macrophage cells used per well or experimental point. The article by Nehls et al. does not indicate this important information in the Materials and methods section, just stating in the main text that “For macrophage infection we treated the cells two hours before infection with Vpx containing VLPs…” (page 7; first top right paragraph). The two experiential conditions used in this work (i.e., scrambled vs siRNA-TDP-43, Figure 6) are nor well controlled and therefore the associated results obtained are not fairly compared and statistically analyzed (n=2).

In this experimental context, we have preferred to avoid the deep analysis of Nehls and coworkers work in our manuscript. However, we have now added an associated paragraph in the Discussion section (page 17, lines 625-641), as follows:

“TDP-43 was first reported as a novel protein that binds to transactivator-responsive DNA sequences within HIV-1, acting as a transcriptional repressor [38]. However, it has been proposed that TDP-43 has no effect on viral LTR transcription and virus production, as in macrophages [39]. In this work, the effect of TDP-43 on p24 release from macrophages was assayed by silencing TDP-43 24 h after infection of Vpx-treated macrophages by HIV/VSV-G-dependent endocytic uptake. Notably, it has been established that HIV-infected macrophages contain large intracellular pools of infectious virus [40-42] which can be released in an exosome-dependent pathway [43] or from late endosomes, where the majority of infectious HIV viral particles assemble and egress [44]. Therefore, it is plausible that under Vpx/VSV-G-mediated viral uptake, p24 expression could be promoted and released from these macrophagic compartments, preventing antiviral TDP-43 action. In permissive non-myeloid cells, HIV-1 viral particles mainly assemble at the cell surface [45-50], as we also reported in virus-producing HEK-293T cells by total internal reflection fluorescence microscopy (TIRFM) [21] and characterized the mechanism of HIV-1 infection promoted by Vif or Nef viral factors and the antiviral action exerted by HDAC6 and/or TDP-43 (TDP-43/HDAC6 axis) in this and other permissive cells [12,19-23,25]. Thus, these data suggest that TDP-43 controls HIV-1 viral production.”

Altogether these data and discussion, we have been assayed the effect of specific TDP-43 silencing by siRNA on HIV-1 infection in HEK-293T and other permissive cell models. Therefore, we have decided to use HIV-1 producing HEK-293T cells, in order to study the effect of TDP-43 knockdown on HIV-1 viral production and virion infection capacity, just allowing the controlled analysis of TDP-43, HDAC6 (their substrates or functions) and Pr55Gag and Vif factors.

  1. In section 2.2, it sounds certain that the decreased level of Gag and Vif proteins are due to the HDAC6 stabilization. However, this was not directly tested in this manuscript. I suggest rephrasing some of the sentences (for example, line 174, "This wt-TDP-43 action on HDAC6 leads to a clearance..., this sounds like it was directly tested).

As suggested by Reviewer #2, we have modified this sentence (lines 174-179), as follows:

“This wt-TDP-43 action on HDAC6 directly correlates with the clearance of the viral proteins Vif and Pr55Gag (Figures 2A, biochemical analysis, lines 3-6, with bands intensity quantification in the histograms on the right, and 2B, western blot line 3 associated bands) as reported for the anti-HIV-1 action of HDAC6 [21,22], observing a TDP-43 dose-response degradative effect on these two viral proteins (Figure 2A) ”.

  1. Related to comment 2, could the authors comment on whether there could be any other explanations for the decreased level of Gag and Vif proteins? For example, could the viral transcription be impaired when overexpressing TDP-43?

We would like to thank Reviewer #2 for arise this remark.

Ou and coworkers showed that TDP-43 overexpression reduces the released of the HIV-1 p24 antigen in HeLa cells co-transduced with HIV-1 proviral DNA (Ou SH, et al. Cloning and characterization of a novel cellular protein, TDP-43, that binds to human immunodeficiency virus type 1 TAR DNA sequence motifs. J Virol. 1995 Jun;69(6):3584-96. doi: 10.1128/JVI.69.6.3584-3596.1995. PMID: 7745706; PMCID: PMC189073). The authors used an in vitro transcription assay to observe that in the presence of recombinant TDP-43, HIV-1-LTR DNA transcription is repressed. However, the activity of TDP-43 to impair HIV-1-LTR DNA transcription is not analyzed in HeLa cells, for example, by quantitatively measuring proviral mRNA production. Although this possibility might be not rule out, the results obtained in our work points into another direction. We observed that the overexpression of functional TDP-43 does not reduce the level of expression of HIV-1 Pr55Gag and Vif proteins, when the autophagy-mediated clearance of Pr55Gag and Vif proteins is inhibited by 3-MA treatment in virus-producing cells. Therefore, TDP-43 does not appear to affect HIV-1-LTR DNA transcription under our experimental conditions. These data suggest that the TDP-43 antiviral action is mainly exerted by regulating the mRNA and enzyme levels of the antiviral factor HDAC6.

We have considered these concepts in a new paragraph, in the Discussion section (lines 642-654), as follows:

“Moreover, it has been reported that TDP-43 overexpression reduces the release of the HIV-1 p24 antigen in HeLa cells cotransduced with HIV-1 proviral DNA [38]. The authors used an in vitro transcription assay to observe that in the presence of recombinant TDP-43, HIV-1-LTR DNA transcription is repressed. However, the ability of TDP-43 to impair HIV-1-LTR DNA transcription has not been analyzed in HeLa cells, for example, by quantitatively measuring proviral mRNA production. Although this possibility might not be ruled out, the results obtained in our work point in another direction. We observed that the overexpression of functional TDP-43 does not reduce the level of expression of HIV-1 Pr55Gag and Vif proteins when the autophagy-mediated clearance of Pr55Gag and Vif proteins is inhibited by 3-MA treatment in virus-producing cells. Therefore, TDP-43 does not appear to affect HIV-1-LTR DNA transcription under our experimental conditions. These data suggest that the anti-HIV-1 action of TDP-43 is mainly exerted by regulating the mRNA and enzyme levels of the antiviral factor HDAC6”.

  1. Figure 3B and 7C, what does this decreased and increased level of full-length Gag imply? Since the level of Gag is normalized to the p24, does this imply that the proteolytic cleavage is more efficient in the case of less infectious virions, and less efficient in the case of more infectious virions?

This is an interesting remark arose by the Reviewer #2.

Our results indicate that TDP-43 negatively affects viral production by reducing the expression level of the Pr55Gag protein. Thus, reduced amount of viral particles were detected in supernatants of cells overexpressing functional TDP-43. The normalization by p24 allows to work with synchronous viral inputs to assay HIV-1 infection capacity, under different experimental conditions. Thus, to obtain synchronous viral amounts, higher supernatant volumes must be taken in TDP-43-overexpressing cells compared to control conductions. The efficiency of the Pr55Gag cleavage could therefore not be compared when p24 amounts are balanced between different experimental conditions. These data also indicate that there are lower amounts of Pr55Gag per virion when produced in cells overexpressing TDP-43.

We feel that to address the question about the efficiency of Pr55Gag cleavage inside virions (i.e., to determine if TDP-43 modulation could affect this proteolytic process) each viral particle produced should content similar amounts of the Pr55Gag protein under each experimental condition. Our results suggest that there are higher amounts of HIV-1 Pr55Gag inside efficient viral particles than inside non-efficient virions. Moreover, the amount of the HIV-1 protease (PR) incorporated per virion should be also considered into account, in order to address this interesting question, thereby requiring al lot of work to be done and other viral models, being out of the scope of this work.

On the other hand, our results regarding the infectious capacity of HIV-1 viral particles correlate with the stability of the Vif viral protein and the amount of this HIV-1 protein incorporated into virions. Noteworthy, these data correlate with previous observations reported by our group that indicate that HIV-1 infectiveness correlates with the stability of Vif, which is inversely dependent on HDAC6 expression level (Valera MS, et al. The HDAC6/APOBEC3G complex regulates HIV-1 infectiveness by inducing Vif autophagic degradation. Retrovirology. 2015 Jun 24;12:53. doi: 10.1186/s12977-015-0181-5. PMID: 26105074; PMCID: PMC4479245). Likewise, it has been reported that HIV-1 infectious activity is directly related to the level of Vif incorporated into nascent virions (Wang Y, et al. HIV-1 Vif inhibits G to A hypermutations catalyzed by virus-encapsidated APOBEC3G to maintain HIV-1 infectivity. Retrovirology. 2014;11:89. doi: 10.1186/s12977-014-0089-5).

We would like to thank Reviewer #2 for all these comments that have improved our work.
